# Deep Learning-Based Automatic Segmentation and Analysis of Mitochondrial Damage by Zika Virus and SARS-CoV-2

**DOI:** 10.3390/v17091272

**Published:** 2025-09-19

**Authors:** Brianda Alexia Agundis-Tinajero, Miguel Ángel Coronado-Ipiña, Ignacio Lara-Hernández, Rodrigo Aparicio-Antonio, Anita Aguirre-Barbosa, Gisela Barrera-Badillo, Nidia Aréchiga-Ceballos, Irma López-Martínez, Claudia G Castillo, Vanessa Labrada-Martagón, Mauricio Comas-García, Aldo Rodrigo Mejía-Rodríguez

**Affiliations:** 1Department of Sciences, Autonomous University of San Luis Potosí (UASLP), San Luis Potosí 78295, Mexico; 2Research Center for Health Sciences and Biomedicine, Autonomous University of San Luis Potosí (UASLP), San Luis Potosí 78210, Mexico; 3Institute of Epidemiological Diagnosis and Reference, InDRE, Mexico City 01480, Mexico; 4School of Medicine, Coordination for the Innovation and Application of Science and Technology, Autonomous University of San Luis Potosí (UASLP), San Luis Potosí 78210, Mexico

**Keywords:** deep-learning, mitochondrial ultrastructure, automated segmentation, SARS-CoV-2, Zika virus

## Abstract

Viruses can induce various mitochondrial morphological changes, which are associated with the type of immune response. Therefore, characterization and analysis of mitochondrial ultrastructural changes could provide insights into the kind of immune response elicited, especially when compared to uninfected cells. However, this analysis is highly time-consuming and susceptible to observer bias. This work presents the development of a deep learning-based approach for the automatic identification, segmentation, and analysis of mitochondria from thin-section transmission electron microscopy images of cells infected with two SARS-CoV-2 variants or the Zika virus, utilizing a convolutional neural network with a U-Net architecture. A comparison between manual and automatic segmentations, along with morphological metrics, was performed, yielding an accuracy greater than 85% with no statistically significant differences between the manual and automatic metrics. This approach significantly reduces processing time and enables a prediction of the immune response to viral infections by allowing the detection of both intact and damaged mitochondria. Therefore, the proposed deep learning-based tool may represent a significant advancement in the study and understanding of cellular responses to emerging pathogens. Additionally, its applicability could be extended to the analysis of other organelles, thereby opening up new opportunities for automated studies in cell biology.

## 1. Introduction

The Severe Acute Respiratory Syndrome Coronavirus 2 (SARS-CoV-2), a member of the *Coronaviridae* family, is an enveloped, positive-sense single-stranded RNA virus ((+)ssRNA) with a genome of approximately 30 kilobases (kb). Numerous viral variants emerged during this global health emergency caused by this virus, exhibiting differences in transmissibility and lethality. However, it remains unclear whether the variation in lethality observed between strains results primarily from the presence of neutralizing antibodies (acquired through vaccination or post-infection), from differences in host–pathogen interactions, or a combination of both [1,2].

The Zika virus (ZIKV), another (+)ssRNA virus, belongs to the *Flaviviridae* family and has a genome of approximately 10.7 kb that encodes three structural proteins—membrane (prM/M), envelope (E), and capsid (C)—and seven non-structural proteins (NS1–NS5). ZIKV was first identified in a rhesus monkey (*Macaca mulatta*) in Uganda in 1947. Mosquitoes of the *Aedes* genus primarily transmit ZIKV, but unlike most flaviviruses, it can also be transmitted sexually and vertically from mother to fetus [3]. Congenital infections are associated with severe outcomes such as microcephaly and Congenital Zika Syndrome (CZS), while adults may experience neurological complications, including meningitis, encephalitis, seizures, and Guillain-Barré syndrome [4].

An almost universal characteristic of (+)ssRNA viruses is that viral replication profoundly remodels the cellular architecture of the producer cell, inducing morphological alterations in organelles as a consequence of host–virus interactions [5]. Thus, these changes can be used to explain the type of antiviral response against a particular pathogen. Therefore, understanding these ultrastructural changes is critical for understanding the viral infection cycle, its pathogenesis, and identifying potential therapeutic targets. It is assumed that a virus can induce either mitochondrial fusion or fission, suggesting that a particular pathogen elicits a predominant immune response across a host population. These are just two examples of how viral infection alters mitochondrial ultrastructure and how these alterations are linked to the immune response triggered during infection. Indeed, mitochondrial fusion has been linked to an anti-inflammatory response, thereby promoting cellular homeostasis. In contrast, fission has been linked to a pro-inflammatory response, facilitating the activation of innate immune signaling pathways [6].

In particular, the type of virus-induced ultrastructural alterations of some organelles can be related to the type of antiviral response. For example, mitochondria are double-membraned organelles essential for cellular energy production by synthesizing adenosine triphosphate (ATP). However, beyond their role in energy metabolism, mitochondria are involved in various cellular processes, including apoptosis, development, cell cycle regulation, and innate immune signaling [6,7]. Mitochondrial morphological alterations are a response to biological insults and immune challenges; morphological processes such as mitochondrial fusion or fission depend on whether the antiviral response is pro- or anti-inflammatory [8].

Recent studies have shown that (+)ssRNA viruses can induce mitochondrial morphological changes associated with distinct immune responses. For instance, Singh et al. reported that ZIKV promotes mitochondrial fission through activation of the NF-κB pathway, whereas SARS-CoV-2 induces mitochondrial fusion, resulting in an interferon-mediated response [6]. Fission refers to dividing mitochondria into smaller organelles, while fusion denotes their integration into a single organelle. It is generally accepted that a given virus predominantly triggers either fission or fusion, suggesting a conserved immune response pattern across infected hosts. These findings exemplify how viral infections reshape mitochondrial ultrastructure in a manner that reflects and potentially drives the host immune response. Therefore, characterizing these mitochondrial alterations could provide insights into the type of immune response elicited (i.e., pro-inflammatory vs. anti-inflammatory), especially when compared to uninfected cells.

In a recent study by Lara-Hernández et al., ultrastructural changes in mitochondria from cell cultures infected with human respiratory syncytial virus (hRSV) were analyzed using transmission electron microscopy (TEM) [9]. However, the analysis was limited to relatively intact mitochondria, introducing a potential selection bias. Moreover, most studies on organelle morphology rely on qualitative descriptions, which are time-consuming and susceptible to observer bias [10]. This work highlighted the fact that a significant challenge in manual TEM analysis is the complexity of the images: multiple subcellular structures with irregular shapes, similar grayscale values, and comparable textures make mitochondria delimitation (segmentation) and classification difficult using standard protocols and techniques. These limitations underscore the need for more accurate and automated analytical tools.

Artificial Intelligence (AI) models, such as artificial neural networks (ANN), can overcome these limitations. In particular, several deep learning-based open-source tools have been developed to study morphological alterations in the mitochondria. For example, MitoMo enables morphological classification and cellular health assessment using fluorescence microscopy [11], while MitoNet was trained using TEM images [12]. Although promising, these tools exhibit limited performance, particularly in pathological contexts, as they are predominantly trained on healthy cells without acquisition protocols that can be widely applied.

Given these limitations, this study aims to develop a deep learning-based tool for the quantitative and qualitative analysis of ultrastructural changes in organelles, particularly mitochondria, in virus-infected cells using TEM imaging. This tool seeks to improve the reliability and statistical power of ultrastructural studies by minimizing subjectivity and enabling the high-throughput analysis of large numbers of images and organelles—tasks that are currently infeasible using manual approaches. The results were highly satisfactory, enabling the automatic identification, segmentation, and analysis of both intact and damaged mitochondria. This significantly reduced processing time and eliminated potential bias.

## 2. Methods and Materials

### 2.1. Generation of TEM Image Database

The SARS-CoV-2 samples, Delta (B.1.617.2) and B.1.1.519 variants, were obtained through a collaboration agreement No. ACOL-2022-DDYR-04 Institute of Diagnosis and Epidemiological Reference (InDRE) and the Center for Research in Health Sciences and Biomedicine (CICSaB/UASLP). The InDRE is a branch of the Mexican Health Ministry (Secretaria de Salud) that by law performs epidemiological surveillance by genotyping, culturing, diagnosing, and storing pathogens that are reported through the Mexican Health system. Therefore, they have the official bank of pathogens identified in Mexico, and thus, through the collaboration agreement No. ACOL-2022-DDYR-04: They grew the SARS-CoV-2 samples at their facilities. Once the infected cultures were chemically fixed and inactivated of any live virus, they were sent to the CICSaB/UASLP for further processing.

The infection of cell cultures was performed in a BSL-3 laboratory, following the methodology described by Harcourt et al. [13] using a Vero cell line. Cultured Vero cells (ATCC, Manassas, VA USA) were prepared for thin-section Transmission Electron Microscopy (TEM) following already established protocols [9,14,15]. The cultured cells were fixed following previous protocols [9,14] in situ.

Ultrathin sections (70 nm thickness) were made using an EM-UC7 ultramicrotome (Leica, Wetzlar, Germany) with a diamond blade (Diatome). The sections were stained with lead citrate and uranyl acetate and observed in a TEM model JEM-JEOL-2100 (JEOL, Tokyo, Japan) at 200 kV, using the objective filter #1 and a OneView Gatan 4K camera, images were taken at 4 kx, 8 kx, 10 kx, 12 kx, 15 kx, 20 kx, and 25 kx magnifications.

In the case of samples infected with the ZIKV, the images used here are part of a dataset generated in Rubio et al. [14]; however, the images used in this project are unpublished.

### 2.2. Selection of Image, Manual Segmentation, and Metric Extraction

For the segmentation of mitochondria in cells infected with SARS-CoV-2, 100 images were selected: 50 corresponding to a negative control (uninfected) and 50 from infected cultures, evenly distributed between Delta (B.1.617.2) (25 images) and B.1.1.519 (25 images) variants. Additionally, 50 images of cells from Rubio et al. [14] were included, consisting of 25 images from control samples and 25 from infected ones.

An image exclusion step was performed manually by an expert observer and applied to all datasets, where images that did not allow a proper observation of large parts of the cell, specifically mitochondria, were discarded. Additionally, images containing myelin bodies, lysosomes, lipid droplets, or multilamellar bodies were excluded to avoid confusion during the neural network training process due to the presence of organelles with similar structural characteristics (see Figure 1). The segmentation and acquisition of mitochondrial metrics were performed manually using ImageJ 1.53a (NIH, Bethesda, MD, USA) [16]. An expert observer reviewed all manual segmentations and mitochondrial metrics. Although this manual exclusion step was needed in the present study, in future work, we will focus on defining a standardized TEM imaging protocol to normalize and homogenize the datasets intended for deep-learning approaches, thereby reducing the need for image exclusion.

A total of 586 and 214 mitochondria were segmented manually for SARS-CoV-2 and ZIKV, respectively. For each image, a binary mask containing the set of manually segmented mitochondria presented on the image was generated, alongside the corresponding binary mask for each mitochondrion. Subsequently, the binary image was multiplied by the original image to isolate the mitochondria from the cellular background.

Three mitochondrial metrics, the Aspect Ratio (AR), Roundness, and Focus of each segmented mitochondria, were determined manually using the software ImageJ, which served as the gold standard for comparison with the metrics obtained automatically. The equations used to calculate each metric are presented below:
(1)
$$AR=\frac{A_{major}}{A_{mior}}$$

(2)
$$Round=\frac{4*A}{\pi *A_{major}^{2}}$$

(3)
$$Focus=\sqrt{A_{major}^{2}-A_{minor}^{2}}$$

where
▪
A=
 Area of the mitochondria▪
$A_{major}=$
 Major axis in the mitochondria▪
$A_{minor}=$
 Minor axis in the mitochondria

### 2.3. Image Preprocessing

The raw images were pre-processed using a Gaussian filter with a standard deviation of 2. This filter decreases the granularity present in the images and reduces noise while maintaining the definition of various cellular structures. This adjustment was used to avoid the loss of fine details, such as mitochondrial cristae, which are essential for the correct segmentation and analysis of mitochondria.

### 2.4. Automatic Segmentation Protocol

The segmentation using Deep-Learning (DL) was based on a convolutional neural network (CNN) with a U-Net architecture, proposed by Ronneberger et al. [17], implemented in Python 3.12.11.

In general, a CNN processes an input image through a series of convolutional layers that progressively identify features of increasing complexity. Initially, simple features such as edges or contrast are detected; as the layers deepen, more complex and abstract patterns are recognized, until complete elements of the image can be distinguished. This process enables the classification of objects within the image and the segmentation of the target structures or the structures of interest [18]. Convolution refers to sliding a filter (kernel) across the original image, computing the dot product between pixels, and producing weighted sums that generate new feature maps, allowing the extraction of relevant characteristics of the object of interest [19]. The activation function acts as a filter that determines which information passes to the next layer, and by modifying or thresholding the output values, activation functions allow the network to learn non-linear relationships and solve more complex problems, while discarding uninformative data. Pooling layers are then used to discriminate among the extracted features, retaining the most dominant ones that will be most useful for object identification and segmentation [18].

The U-net architecture itself consists of two symmetric parts. The left side, known as the encoder, focuses on feature extraction through 4 blocks. Each block comprises two 3 × 3 convolutional layers with a Rectified Linear Unit (ReLU) activation function, followed by a 2 × 2 Maxpooling layer that downsample the feature maps, reducing their spatial dimensions while retaining the most salient information. This process allows the network to extract increasingly abstract and relevant features from the regions to be segmented. The right side of the architecture, referred to as the decoder, consists of 4 blocks with transposed convolutions (also known as up-sampling or deconvolution layers). These layers progressively expand the feature maps to their original spatial dimensions, reconstructing the image. During this process, the decoder leverages the features extracted by the encoder to perform the segmentation. In summary, CNNs enable hierarchical feature extraction, and the U-Net architecture applies this principle through contraction on the left side to extract features and expands on the right side to reconstruct the segmented output (see Figure 2).

To improve the performance of the proposed DL tool, data augmentation was implemented, thereby enriching the available image set. The network was trained with input images of original dimension 4096 × 4096 pixels, which were resized to 256 × 256 pixels to reduce computational time during training. The network was trained for 100 epochs, where an epoch is defined as one complete training cycle over the dataset. Initially, the training set consisted of 80 images, and a batch size of 10 random images was used. Each input image underwent random geometric transformations, including rotation, horizontal and vertical translation, zooming in on the image, and horizontal flipping; not all transformations were necessarily applied to each image.

Subsequently, the empty spaces generated by these transformations were filled using the neighboring pixel values, aiming to improve training and prevent overfitting of the network. Furthermore, it was established that each epoch would consist of 30 steps, meaning each epoch would be composed of 30 batches of transformed images. This resulted in a total of 300 geometrically transformed images per epoch, which, when repeated over 100 epochs, led to a training dataset of 30,000 images.

### 2.5. Validation and Statistical Analyses

Following automatic segmentation, each mitochondrion was independently analyzed to extract quantitative morphological metrics. This automated analysis was performed using MATLAB^®^ 2021, enabling the systematic estimation of parameters (i.e., AR, Roundness, and Focus) for each organelle.

In addition, a global performance of the tool was evaluated by overlapping automatic and manual segmentation of each TEM image using the DICE similarity coefficient (DSC) [20]. This index ranges from 0 to 1, where a value of 1 indicates perfect overlap between the compared segmentations, while a value of 0 indicates no similarity, and is defined as
(4)
$$DSC=\frac{2\left|A\cap B\right|}{\left|A\right|+\left|B\right|}$$

where

A, B = Set of pixels that conform to a binary mask|.| = Cardinality of each set

The normality of the data was determined using a Kolmogorov–Smirnov test, which showed that the data did not follow a normal distribution. Consequently, the non-parametric Kruskal–Wallis test with Dunn’s *post hoc* was applied to assess whether there were significant differences between the magnifications used, the types of viruses, and their potential impact on the tool’s performance for the automatically segmentation. Additionally, Lin’s Concordance Correlation Coefficient (CCCLin) [21] was evaluated using R 4.2.1^®^ [22]. This method allows for comparing two measurements of the same variable, using an existing measurement technique as the gold standard (manual metrics measurements). CCCLin is defined as:
(5)
$$\rho_{c}=\frac{2\rho \sigma_{x}\sigma_{y}}{{\left({µ}_{x}-{µ}_{y}\right)}^{2}+\sigma_{x}^{2}\sigma_{y}^{2}}$$

where

x = Dataset as the gold standardy = Dataset to be evaluatedρ = Correlation coefficient between sets

The value of ρ_c_ ranges from −1 to 1, with perfect agreement when the values approach 1, perfect discordance when they approach −1, and no agreement when it is 0.

All images used in this study were processed in Google Colab Pro, on a Lenovo computer equipped with an Intel Core i7 (OR, USA) processor and an NVIDIA GeForce GTX GPU (CA, USA), enabling efficient model execution and the handling of large volumes of data.

## 3. Results

### 3.1. Manual Data Acquisition

A total of 586 mitochondria were segmented in images of cells, including SARS-CoV-2-infected and negative control samples, while 214 mitochondria were segmented in images of cells, including ZIKV-infected and negative control samples. It is important to note that the cell lines used in this study exhibit both damaged and healthy mitochondria, even in the non-infected samples. Therefore, the segmentation was performed considering both observed morphologies, depending on the cell section, to closely approximate the original morphology of the mitochondria and minimize measurement errors, ensuring the inclusion of all mitochondria regardless of their functional state. A representative example of the images used, and their respective manual segmentations, is shown in Figure 3.

Table 1 summarizes the number of electron images analyzed and the corresponding count of individually segmented mitochondria. Table 2 shows the means and standard deviations of the metrics obtained manually, which were used as reference values for comparison with the metrics obtained automatically.

Figure 4 and Figure 5 show some of the morphometric differences in mitochondria resulting from cell-virus interaction. In the case of SARS-CoV-2, mitochondria infected with the B.1.1.519 variant exhibit a higher AR (Figure 4a), lower values of Roundness (Figure 4b), and higher Focus values (Figure 4c) compared to mitochondria infected with the Delta (B.1.617.2) variant and the control group (Figure 4b). These results suggest that both SARS-CoV-2 variants affect the mitochondria differently. Infection with B.1.617.2 generates less elongated mitochondria, while infection with B.1.1.519 results in elongated mitochondria. These results suggest that these variants have a considerably different effect on this organelle and possibly on the immune response against these viruses.

As expected, Figure 5 shows that, in general, the morphology of the mitochondria is altered by the ZIKV infection. However, the effect of this viral infection was different from that caused by both SARS-CoV-2 variants. The AR and the Focus of the mitochondria of ZIKV-cells decrease with respect to uninfected cells, while the Roundness increases, suggesting that the mitochondria become less elongated.

### 3.2. Automatic Segmentation

The neural network training was performed using images to which a Gaussian filter was applied, yielding favorable automatic segmentation results (see Figure 6 and Figure 7). This tool demonstrated highly accurate performance in the automatic delimitation of mitochondria in TEM images of cells infected with SARS-CoV-2 and ZIKV. For the SARS-CoV-2 dataset, the model achieved an accuracy and loss of 0.991 and 0.021, respectively, while for the ZIKV dataset, the corresponding values were 0.995 and 0.011. These results indicate that the model correctly identifies mitochondrial structures from the background in both viral infection contexts.

Figure 6b,c and Figure 7b,c show that the mitochondria were correctly identified in both cells infected with ZIKV and cells infected with the Delta (B.1.617.2) and B.1.1.519 variants. This was accomplished despite the various complex morphologies present in the analyzed images. However, in cases where the mitochondria are very close to each other, the automatic segmentation presents some difficulties, leading to artificial fusion of some of them. Nonetheless, the results obtained with this methodology are considered satisfactory, as proper identification was achieved in most cases.

### 3.3. Automatic Metric Extraction

To perform the automatic measurement of metrics of interest, only mitochondria with a well-defined individual segmentation were considered (see Table 3); this led us to use only 50% of mitochondria for SARS-CoV-2 samples and 68% of mitochondria for the ZIKV samples. A total of 447 individually segmented mitochondria were used to automatically obtain metrics for evaluating the tool and comparing the results with manual estimates.

For the mitochondria individually segmented using the automated U-Net model, the resulting morphometric metrics were consistent with those obtained through manual annotation. Table 4 presents the mean and standard deviation values of the AR, Roundness, and Focus. A comparative evaluation of mitochondrial morphometric parameters obtained via manual and automatic segmentation revealed a high consistency between both techniques across all experimental groups. That is, there are no statistical differences between the data obtained manually and automatically.

Figure 8 and Figure 9 summarize the distributions of AR, Roundness, and Focus of mitochondria from cells infected with SARS-CoV-2 variants, ZIKV, and their respective control groups. While statistically significant differences were observed between infections, no significant differences were found between the segmentation techniques (manual and automatic) within the same group. These results show that the AI-based automatic segmentation pipeline reliably reproduces morphological features and metrics of mitochondria, even in the presence of virus-induced alterations, supporting its validity for quantitative analysis in transmission electron microscopy datasets.

### 3.4. Validation and Statistical Test

To evaluate the tool, the DSC was calculated to globally compare the obtained segmentations by overlaying the automatic segmentation image with the manual segmentation image, and assessing the similarity between them, regardless of the individual segmentation of each mitochondrion. The data was analyzed using the Kolmogorov–Smirnov test to determine that the data did not follow a normal distribution. Consequently, to evaluate statistical differences between the metrics obtained, the non-parametric Kruskal–Wallis test was used.

Comparison of the segmentation results considering the magnification factor used during TEM image acquisitions for SARS-CoV-2 and ZIKV is presented in Table 5 and Table 6. The results showed significant differences (*p* < 0.05) between the magnification and virus-type groups in the evaluated variables (AR, Roundness, and Focus); however, and more importantly, no significant differences were observed when comparing the manually obtained results with those obtained automatically using the proposed deep learning approach (see Figure 10 and Figure 11). In particular, for SARS-CoV-2, the results of AR and Roundness suggest that this value depends on the magnification, while Focus is not affected by this factor. On the other hand, for ZIKV results, no metric was affected by the magnification.

The CCCLin was employed to compare the metrics obtained manually, considered the gold standard, with those obtained automatically. It is important to note that for this validation, only the data from mitochondria with a well-defined individual automatic segmentation were considered. Table 7, Table 8 and Table 9 present the values obtained from the CCCLin for each of the metrics calculated for the mitochondria in the control and infected groups. The statistical analysis showed no differences between the data obtained manually and automatically in these tables.

To evaluate the automatic segmentations of the whole images, the DSC was calculated for both datasets. A histogram was also plotted for DSC values, with ranges between 0.09 and 0.99. For SARS-CoV-2 images (Figure 12), the fact that 85% of the images have a DSC equal to or greater than 0.79 indicates that most mitochondria were correctly identified during the automatic segmentation process. For ZIKV images (Figure 13), 84% of the images have a DSC equal to or greater than 0.80, indicating again that most mitochondria were segmented correctly using the proposed automatic approach. These results indicate that, overall, for each image, the automatic segmentation is quantitatively very similar to the corresponding manually performed segmentation, despite the presence of some mitochondria that appear to be fused.

## 4. Discussion

In this study, we developed an AI-based tool to identify, segment, and characterize mitochondria from infected cultured cells with either SARS-CoV-2 or ZIKV, utilizing a deep-learning approach with a convolutional neural network.

First, from the point of view of image analysis, preprocessing of TEM images applying a Gaussian filter with a σ value of 2 was sufficient to improve the automated segmentation process. The removal of noise and image smoothing facilitated the neural network’s ability to learn the features of the mitochondria better, yielding results comparable to those achieved by manual identification by independent experts; the DSC was greater than 0.85 in terms of overall segmentation similarity, indicating high accuracy in automated segmentation compared to manual segmentation. Although the segmentation did not perfectly delineate each mitochondrion, our deep learning system was able to identify intact, damaged, and partially visible mitochondria. The correlation coefficient (CCCLin) between the metrics obtained automatically and manually had an average of 0.94, highlighting the model’s effectiveness.

Despite the challenges of working with thin-section TEM images, which contain numerous organelles with similar morphologies, variations in grayscale intensities, and diverse textures, our tool achieved highly satisfactory results. It is important that automated analysis of thin-section TEM images is considerably more complex than segmentation studies based on fluorescence images, such as MitoMo [11], where specific markers and clearer contrast facilitate structure detection.

In the context of 2-D electron microscopy (EM) data, several approaches for mitochondrial segmentation have been reported. Traditional image processing methods, such as thresholding or edge-detection algorithms, as well as semi-automatic annotation tools like Ilastik [23], have been widely used. However, these strategies generally require manual correction and often perform poorly when images present strong noise, overlapping structures, or heterogeneous textures. More recent, deep learning-based methods for 2-D EM segmentation (e.g., MitoNet [12]) have improved accuracy and reduced the need for manual intervention. Nevertheless, most of these approaches have been trained on datasets containing only morphologically intact mitochondria, which limits their applicability in pathological contexts, such as viral infections.

Our work addresses this limitation by training the model with both mock-treated controls (non-pathological conditions) and infected cells, explicitly incorporating mitochondria with severe morphological alterations. This strategy represents a significant improvement over previous studies [9,12] that relied solely on morphologically intact mitochondria, which may limit the applicability of their findings to pathological contexts. The inclusion of images with varying degrees of mitochondrial damage provides greater robustness and generalizability to the tool. Nonetheless, we still need to standardize a protocol for image acquisition to further enhance the model’s performance by reducing image variability and improving input consistency.

From a biological point of view, this preliminary study has been extremely helpful in gaining insight into the effect of viral infection in cultured cells. During the viral replication cycle, the reconfiguration of cellular metabolism plays a pivotal role in the survival of viruses. Viruses, such as SARS-CoV-2 and ZIKV, have intricate mechanisms to manipulate host cell functions in their favor, ultimately evading the immune response [24,25]. In this context, the role of mitochondria has been investigated in the life cycle and cell death process, particularly in response to viral infection.

It has been shown that mitochondrial function and structure undergo alterations during viral infection, disrupting pathways such as apoptosis and the immune system activation mediated by mitochondria [26]. On the one hand, SARS-CoV-2 infection has been studied at both the molecular and ultrastructural levels to elucidate mitochondrial alterations. On the other hand, there remains a lack of information regarding the ultrastructural modifications observed during infection. Furthermore, ultrastructural studies focusing on alterations in mitochondrial dynamics of different variants of SARS-CoV-2 are even more limited [27].

Here, we compared the ultrastructural modifications in mitochondria caused by the infection of two variants of SARS-CoV-2: the Delta variant (B.1.617.2) and B.1.1.519. To achieve this, we employed a deep learning approach with a convolutional neural network to segment mitochondria in images. Initially, each mitochondrion was manually segmented, focusing on its outer membrane and cristae. Subsequently, we classified the mitochondria into two groups: healthy mitochondria with no morphological alterations and damaged mitochondria with cristae and outer membrane disorders. These methodologies have been extensively used in patient biopsies, in vitro models, and tissues from murine models, which have shown that mitochondrial morphology changes during infection, leading to swelling and vacuolization [28,29]. In addition to these studies, we observed alterations in mitochondrial swelling and vacuolization at different stages.

To perform an exhaustive analysis of the ultrastructural alterations, size parameters, and shape descriptors were selected for analysis within each mitochondrion. Among the parameters describing changes in mitochondrial morphology during SARS-CoV-2 infection are the size and density of mitochondria [30]. Additionally, size parameters, including the area, external perimeter, and Feret diameter, as well as shape descriptors such as form factor, circularity, and Roundness, have been analyzed [31]. It is important to point out that quantitative measures to characterize the morphological changes in mitochondria, particularly in stress conditions or in response to certain stimuli, such as surface area, perimeter, roundness, aspect ratio, and form factor, have been evaluated by others [32,33]. On the one hand, changes in roundness indicate fragmentation and cellular stress, enabling the direct description of mitochondria with circular morphologies. On the other hand, the aspect ratio is used to describe the elongation of organelles, suggesting an elongated morphology [33]. While the shape of the mitochondria and the assembly of mitochondrial networks have been primarily associated with fission and fusion processes, the modification of mitochondrial architecture remains an area of opportunity [34]. Some studies have utilized fluorescence microscopy to observe the formation of mitochondrial networks and describe the shapes of mitochondria associated with fission and fusion events [35]. However, this technique’s resolution limits its ability to quantify the level of damage sustained at the ridges and mitochondrial membrane, which is crucial for understanding the factors that can alter mitochondrial architecture. Therefore, electronic microscopy is an essential technique for comprehending these modifications [36]. Nevertheless, the development of artifact-free preparation methods for evaluating mitochondrial morphological changes under experimental conditions is crucial. Several factors should be considered, such as the choice of fixing reagent and the appropriate fixing time, to preserve ultrastructure and minimize the formation of artifacts [37]. It has been established that the use of insulin as a preservative agent can enhance mitochondrial fusion processes [38]. The chemical method employing aldehydes, particularly glutaraldehyde, remains the most prevalent approach for studying cell ultrastructure. It is essential to note that the fixation time can lead to the formation of artifacts that may adversely impact the ultrastructure negatively [39]. High-pressure freezing and freeze-substitution techniques offer the potential to quickly stop cell activity, thereby mitigating structural artifacts that commonly arise during chemical fixation [40]. The high cost of high-pressure freezing and freezing substitution techniques makes chemical fixation the most commonly employed and relevant method due to its continued necessity, especially when the aim of the study is not a 3D reconstruction [41]. In our research group, the previously described chemical fixation method has been utilized, facilitating the study of the ultrastructure of diverse cell lines with a relatively low number of artifacts [14,42,43].

By analyzing some of these shape descriptors, we observed that mitochondria infected with the B.1.1.519 variant exhibited a larger AR and low Roundness values, along with high Focus values compared to the Delta variant (B.1.617.2) and the control group. This suggests that mitochondria of cells infected with the B.1.519 variant are more elongated than those infected with the Delta variant. This suggests that infection with B.1.519 results in mitochondrial fusion while Delta, most likely, results in mitochondrial fission. The lower correlation in AR measurements for the Delta (B.1.617.2) variant could be attributed to the more aggressive or damaging pathology of this variant, which likely alters mitochondrial morphology more significantly and thus impairs automated detection.

SARS-CoV-2 Delta (B.1.617.2) has been consistently associated with more severe clinical outcomes compared to other SARS-CoV-2 variants. Multiple cohort studies have demonstrated that infection with the Delta (B.1.617.2) variant increases the severity of disease compared to B. 1.1.7 [44,45,46]. This suggests a higher burden of cellular and subcellular stress—including mitochondrial damage—which could reduce the consistency between manual and automated measurements. Therefore, it is likely that the diminished correlation observed for AR likely reflects variant-specific mitochondrial disruption in Delta (B.1.617.2). These findings align with the results of a study conducted on the mitochondria of monocytes from patients diagnosed with COVID-19. The study revealed significantly higher AR and lower Roundness values [31].

The mitochondrial fusion process entails the fusion of individual mitochondria within a single organelle, resulting in a more elongated morphology. This process is facilitated by the fusion of the outer mitochondrial membrane (OMM), mediated by mitofusin 1 (MFN1) and mitofusin 2 (MFN2), and the fusion of the inner mitochondrial membrane (IMM) is mediated by optic atrophy 1 (OPA1) [47]. Mitochondrial fusion initiates the signaling pathway of the RIG-I (RLR) receptors, which can detect viral RNA in the cytoplasm and subsequently activate the mitochondrial antiviral signaling protein (MAVS). MAVS, in turn, triggers the production of interferons (IFNs), which are pivotal proteins in the defense against viruses. Consequently, mitochondrial elongation enhances innate antiviral immunity mediated by RIG-I, which is induced by viral infection and inhibits viral replication [6]. This phenomenon may be associated with the severity of the disease during the progression of infection by the B.1.1.7 variant. In Mexico, the second wave of infection was predominantly characterized by the B.1.1.519 variant, which was confined to the country and not globally disseminated. This wave marked the highest peak of mortality rates [48,49]. This observation can be attributed to the ultrastructural modifications caused by this variant and the subsequent induction of an intensified immune response. However, our study did not focus on the signaling pathways mediated by mitochondrial dynamics dysfunction. Consequently, further research is needed to corroborate these findings.

In contrast, mitochondrial fission is a highly regulated dynamic process that results in the formation of two or more smaller mitochondria. The primary protein responsible for mediating mitochondrial fission is dynamin-related protein 1 (DRP1). Certain viruses, such as hepatitis viruses, can induce the propensity towards mitochondrial fission, leading to the suppression of the innate immune response, apoptosis, and enhancing viral replication [50]. This induction behavior of fragmented mitochondria with smaller sizes is observed in our study in the infected cell with the Delta (B.1.617.2) variant. Specifically, we observed a low AR value, high Roundness values, and low Focus values, which suggests a less elongated and more rounded morphology of mitochondria. The induction of mitochondrial fission during infection by the Delta variant may be linked to its rapid dissemination, which is twice as rapid as other variants [51]. However, further research is necessary to elucidate the reasons behind the modified mitochondrial dynamics, as observed in different variants. While some variants favor mitochondrial fusion, others favor fission, as evidenced by the data reported in this study.

To date, only one study has comprehensively described the changes induced by various SARS-CoV-2 variants of interest, revealing morphological alterations in infected cells. Notably, the Delta variant induces ultrastructural swelling, mitochondrial vacuolization, and disorganization of mitochondrial cristae. Although the most prevalent changes observed in this study were confined to mitochondria, regardless of the kinetics time and the infection variant, it is acknowledged that the virus can cause mitochondrial damage through diverse pathways and strategies [27].

The same segmentation tool was used to analyze ultrastructural modifications in a cell culture infected with the ZIKV. Infected cells had lower shape descriptor values, such as the AR and Focus, as well as higher Roundness values, than the control samples. This suggests that ZIKV infection, like the Delta variant, modifies mitochondrial morphology, resulting in less elongated mitochondria and the activation of mitochondrial fission. This aligns with previous studies showing ZIKV infection induces mitochondrial fragmentation through the negative regulation of MFN2 expression, reducing mitochondrial fusion capacity [52].

From a functional perspective, developing tools like deep learning with a convolutional neural network to segment mitochondria in images is crucial for the rapid detection of changes during infection and an initial approach to understanding the progression of an infection without having to use methods that require reagents specific to a particular pathogen. This enables novel experiments to elucidate signaling pathways induced in infections and the reasons behind a variant’s or a certain virus’s preference for inducing mitochondrial fission or fusion. Furthermore, automated segmentation dramatically reduced analysis time, from months of manual work to just 30 min for a complete dataset segmentation and enabled the rapid and objective extraction of morphological metrics. This enables the identification of damaged mitochondria and facilitates the study of mitochondrial alterations in various viral contexts.

Finally, our tool also provides an adaptable platform for investigating other subcellular structures in an objective and high-throughput manner. Its application could be extended to the analysis of other organelles affected by viral infections, opening new avenues for the automated study of cellular biology.

## 5. Conclusions

In the present work, we developed a deep learning-based tool for analyzing TEM images, aimed at automatically segmenting and characterizing mitochondria in cells infected with two different SARS-CoV-2 variants and ZIKV. The results obtained were highly satisfactory, achieving an accuracy greater than 85% in the identification, segmentation, and metrics extraction of mitochondria. This tool significantly reduces processing time and minimizes the subjectivity of manual morphological studies of cellular architecture. Additionally, this preliminary approach enables a prediction of the immune response to viral infections by allowing the detection of both intact and damaged mitochondria, thus eliminating potential biases in the results, despite the absence of a standardized protocol for image acquisition and the variability in magnification and virus-induced structural damage. Nonetheless, albeit in a preliminary state, this approach allowed us to propose a cellular response against two different SARS-CoV-2 variants and ZIKV.

Our AI-based tool represents a significant advancement in the study of mitochondrial remodeling in the context of viral infections, with important implications for understanding cellular responses to emerging pathogens such as SARS-CoV-2 and ZIKV. Furthermore, its applicability could be extended to the analysis of other organelles, opening new opportunities for automated studies in cell biology.

## Figures and Tables

**Figure 1 viruses-17-01272-f001:**
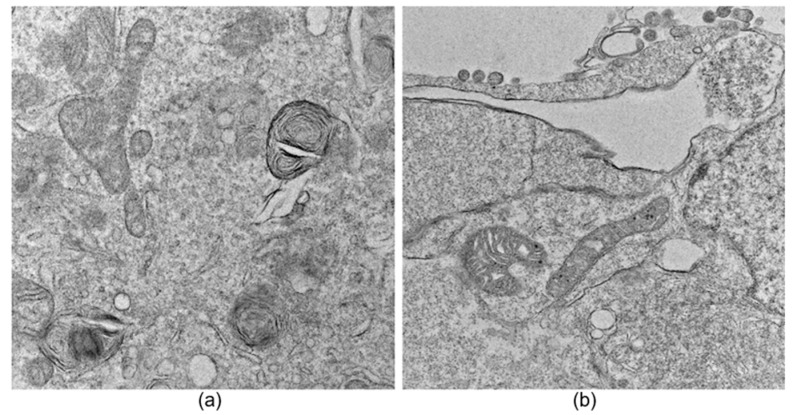
Thin-section TEM Image. (**a**) Example of an image not used to train the network due to the variety of organelles observed; (**b**) representative example of an image used for network training [14].

**Figure 2 viruses-17-01272-f002:**
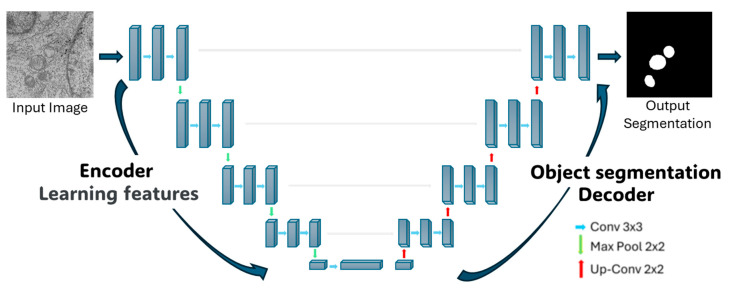
Representation of the U-Net architecture used for automatic segmentation of mitochondria. The U-net uses two symmetric sides, with a common convolutional neural network (CNN) architecture on the left side, and transposed convolutions on the right side to expand the image.

**Figure 3 viruses-17-01272-f003:**
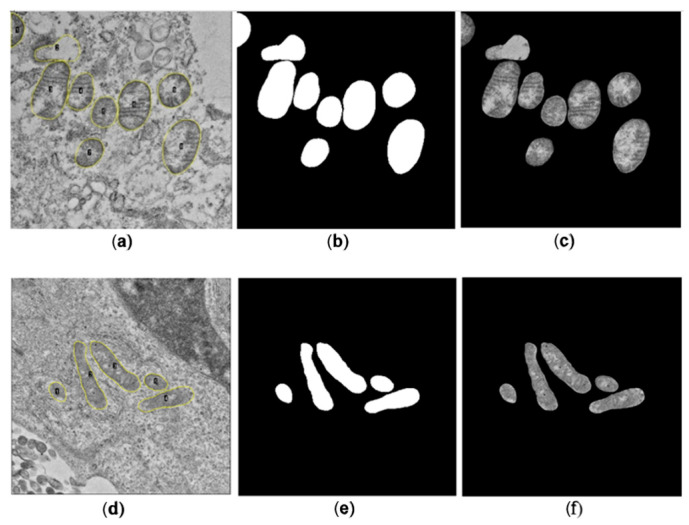
Example of mitochondria manual segmentation for representative thin-section TEM images. (**a**) TEM image of Vero cells (negative control for the SARS-CoV-2 group), (**b**) segmented mitochondria from panel represented as binary masks, (**c**) overlap of images in panel a and b. Panels (**d**–**f**) represent the same process applied to an uninfected (negative control) astrocyte for the ZIKV group.

**Figure 4 viruses-17-01272-f004:**
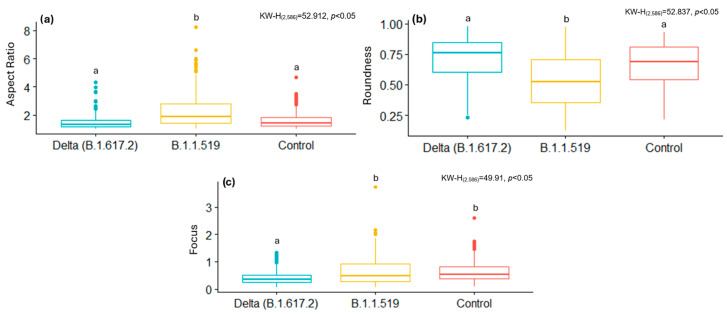
Box plots of the metrics obtained manually from mitochondria in control and SARS-CoV-2-infected cells. (**a**) Aspect Ratio, (**b**) Roundness, (**c**) Focus. Normality tests were performed using Kolmogorov–Smirnov, and, for comparisons between groups, the Kruskal–Wallis test was used with a Dunn *post hoc* analysis with different letters indicating significant statistical differences (*p* < 0.05).

**Figure 5 viruses-17-01272-f005:**
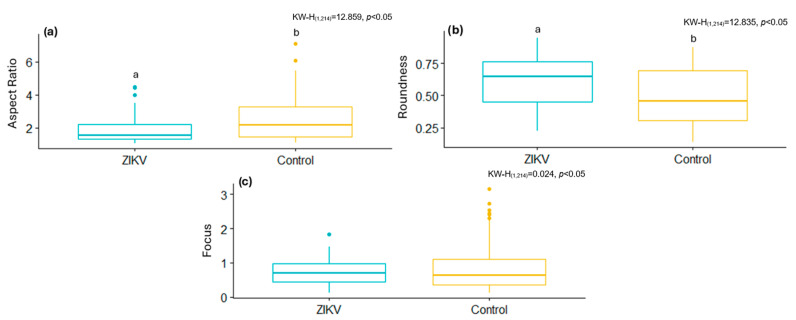
Box plots of the metrics obtained manually from mitochondria in control and ZIKV-infected cells. (**a**) Aspect Ratio, (**b**) Roundness, (**c**) Focus. Normality tests were performed using Kolmogorov–Smirnov, and, for comparisons between groups, the Kruskal–Wallis test was used with a Dunn *post hoc* analysis with different letters indicating significant statistical differences (*p* < 0.05).

**Figure 6 viruses-17-01272-f006:**
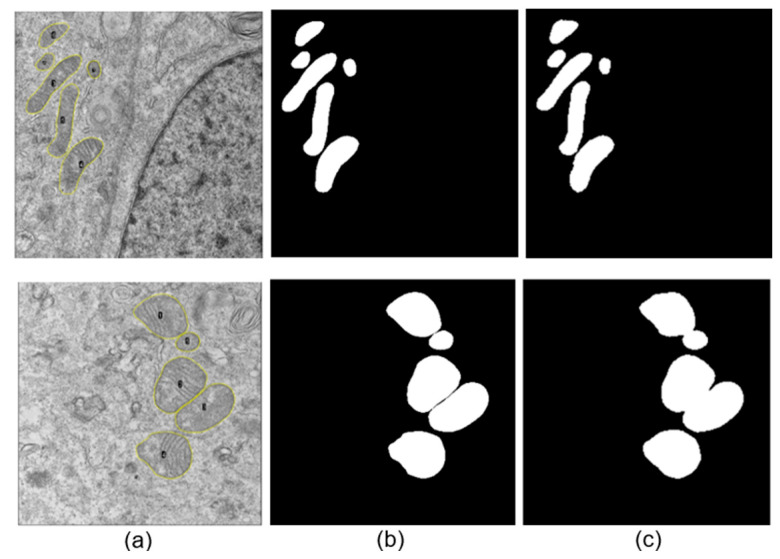
Examples of automatic segmentations of mitochondria for images of cells infected with ZIKV after adding a Gaussian filter. (**a**) Original images of cells infected with ZIKV, (**b**) manually performed segmentation, (**c**) automatic segmentations.

**Figure 7 viruses-17-01272-f007:**
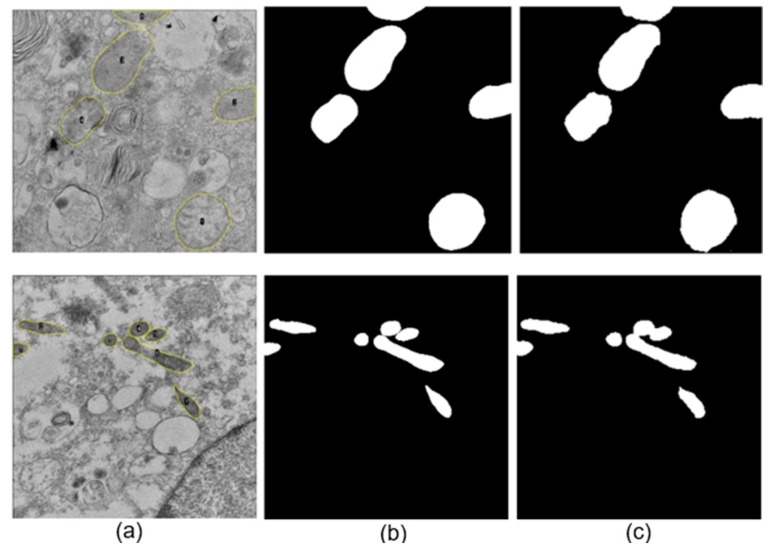
Examples of automatic segmentations of mitochondria for images of cells infected with SARS-CoV-2 after adding a Gaussian filter. (**a**) Original images of cells infected with SARS-CoV-2, (**b**) manually performed segmentation, (**c**) automatic segmentations. The upper row corresponds to a cell infected with Delta (B.1.617.2), and the lower row to a cell infected with B.1.1.519.

**Figure 8 viruses-17-01272-f008:**
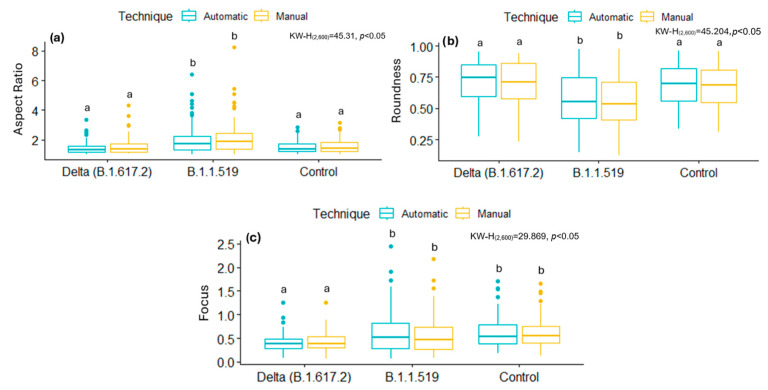
Box plots comparing metrics obtained manually and automatically in control and SARS-CoV-2 infected images, displaying mean and percentiles. (**a**) Aspect Ratio, (**b**) Roundness, (**c**) Focus. Normality tests were performed using Kolmogorov–Smirnov, and, for comparisons between groups, the Kruskal–Wallis test was used with a Dunn *post hoc* analysis with different letters indicating significant statistical differences (*p* < 0.05).

**Figure 9 viruses-17-01272-f009:**
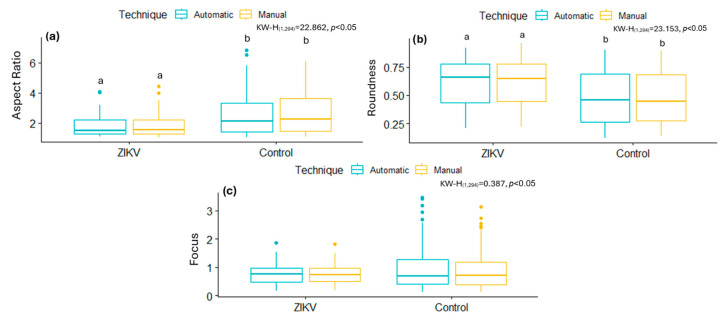
Box plots comparing the metrics obtained manually and automatically in control and ZIKV-infected images, displaying mean and percentiles. (**a**) Aspect Ratio, (**b**) Roundness, (**c**) Focus. Normality tests were performed using Kolmogorov–Smirnov, and, for comparisons between groups, the Kruskal–Wallis test was used with a Dunn *post hoc* analysis with different letters indicating significant statistical differences (*p* < 0.05).

**Figure 10 viruses-17-01272-f010:**
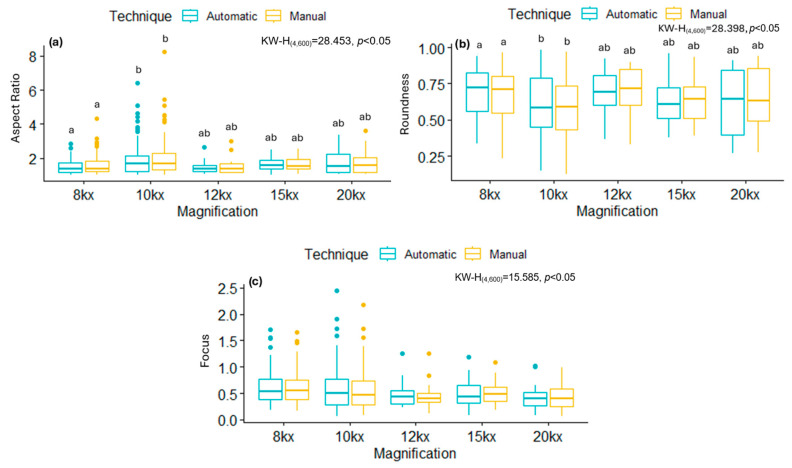
Box plots comparing magnifications used relative to the metric obtained and the technique employed in control and SARS-CoV-2-infected images. (**a**) Aspect Ratio, (**b**) Roundness, (**c**) Focus. Normality tests were performed using Kolmogorov–Smirnov, and, for comparisons between groups, the Kruskal–Wallis test was used with a Dunn *post hoc* analysis with different letters indicating significant statistical differences (*p* < 0.05).

**Figure 11 viruses-17-01272-f011:**
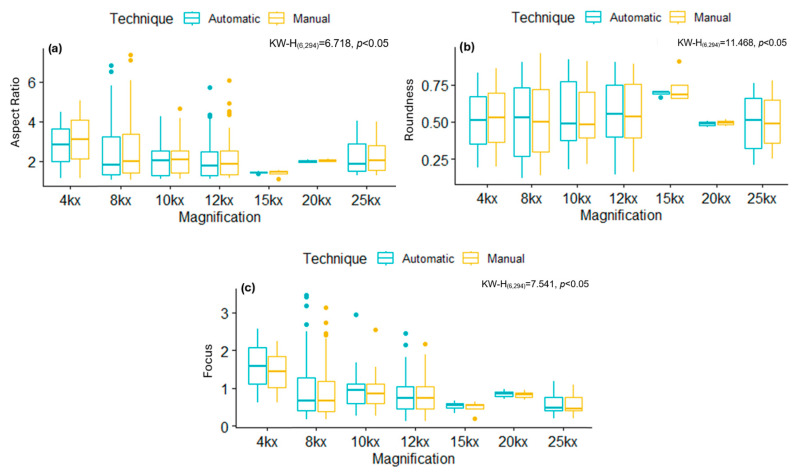
Box plots comparing magnifications used relative to the metric obtained and the technique employed in control and ZIKV-infected images. (**a**) Aspect Ratio, (**b**) Roundness, (**c**) Focus. Normality tests were performed using Kolmogorov–Smirnov, and, for comparisons between groups, the Kruskal–Wallis test was used with a Dunn *post hoc* analysis with different letters indicating significant statistical differences (*p* < 0.05).

**Figure 12 viruses-17-01272-f012:**
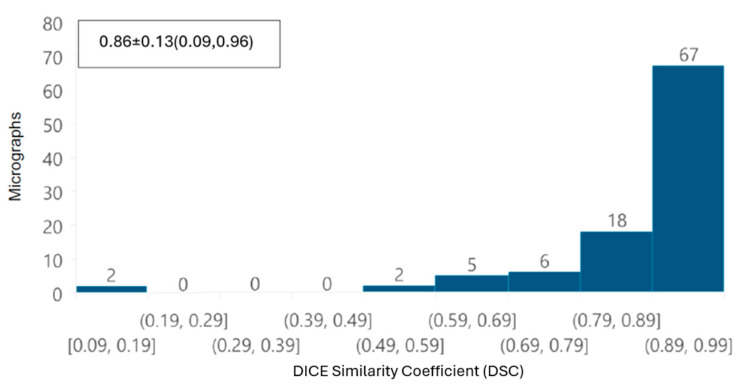
DSC histogram for each image of control cells and cells infected with Delta (B.1.617.2) and B.1.1.519 variants, with mean ± standard deviation (min, max) values of 0.86 ± 0.13 (0.09,0.96) were found.

**Figure 13 viruses-17-01272-f013:**
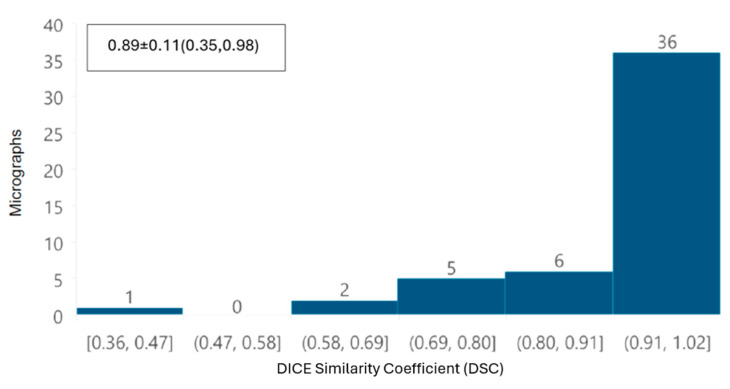
DSC histogram for each image of control cells and cells infected with ZIKV, mean ± standard deviation (min, max), with mean ± standard deviation (min, max) values of 0.89 ± 0.11 (0.35,0.98) were found.

**Table 1 viruses-17-01272-t001:** Total number of manually segmented mitochondria in images of cells infected with SARS-CoV-2 and ZIKV.

		Image	# Individual Mitochondria	Total
**SARS-CoV-2**	Delta (B.1.617.2)	25	142	586
	B.1.1.519	25	151	
	Control	50	293	
**ZIKV**	Infected	25	86	214
	Control	25	128	

**Table 2 viruses-17-01272-t002:** Average and standard deviation metrics for manually segmented mitochondria in control cells and cells infected with SARS-CoV-2 and the ZIKV.

		Aspect Ratio	Roundness	Focus
**SARS-CoV-2**	Delta (B.1.617.2)	1.55 ± 0.61	0.71 ± 0.18	0.41 ± 0.26
	B.1.1.519	2.34 ± 1.23	0.54 ± 0.21	0.66 ± 0.53
	Control	1.62 ± 0.55	0.67 ± 0.16	0.63 ± 0.36
**ZIKV**	Infected	1.86 ± 0.81	0.61 ± 0.20	0.73 ± 0.36
	Control	2.53 ± 1.35	0.50 ± 0.23	0.83 ± 0.62

**Table 3 viruses-17-01272-t003:** Total number of manually and automatically segmented mitochondria, with the percentage of mitochondria obtained automatically with respect to those segmented manually.

		Manually Segmented Mitochondria	Total Mitochondria	Automatically Segmented Mitochondria	Total Mitochondria
**SARS-CoV-2**	Delta (B.1.617.2)	142	586	50 (35%)	300 (51%)
	B.1.1.519	151		102 (67%)	
	Control	293		148 (50%)	
**ZIKV**	Infected	86	214	57 (66%)	147 (68%)
	Control	128		90 (70%)	

**Table 4 viruses-17-01272-t004:** Average and standard deviation metrics for automatically segmented mitochondria.

		Aspect Ratio	Roundness	Focus
**SARS-CoV-2**	Delta (B.1.617.2)	1.48 ± 0.48	0.71 ± 0.17	0.42 ± 0.22
	B.1.1.519	1.99 ± 0.97	0.57 ± 0.22	0.62 ± 0.44
	Control	1.51 ± 0.37	0.69 ± 0.16	0.62 ± 0.31
**ZIKV**	Infected	1.78 ± 0.71	0.62 ± 0.20	0.78 ± 0.36
	Control	2.53 ± 1.36	0.50 ± 0.24	0.83 ± 0.81

**Table 5 viruses-17-01272-t005:** Kruskal–Wallis test for each metric in control and infected SARS-CoV-2 images with Delta (B.1.617.2) and B.1.1.519 variants.

SARS-CoV-2		
		*p*-Value (<0.05)
Aspect ratio	Magnification	1.009 × 10^−5^
	Groups (control vs. infected)	1.449 × 10^−10^
	Technique (manual vs. deep-learning)	0.1224
Roundness	Magnification	1.036 × 10^−5^
	Groups (control vs. infected)	1.528 × 10^−10^
	Technique (manual vs. deep-learning)	0.523
Focus	Magnification	0.003629
	Groups (control vs. infected)	3.266 × 10^−7^
	Technique (manual vs. deep-learning)	0.5643

**Table 6 viruses-17-01272-t006:** Kruskal–Wallis test for each metric in control and ZIKV-infected images.

ZIKV		
		*p*-Value (<0.05)
Aspect ratio	Magnification	0.3476
	Groups (control vs. infected)	1.74 × 10^−6^
	Technique (manual vs. deep-learning)	0.4358
Roundness	Magnification	0.2737
	Groups (control vs. infected)	1.496 × 10^−6^
	Technique (manual vs. deep-learning)	0.971
Focus	Magnification	0.07493
	Groups (control vs. infected)	0.5337
	Technique (manual vs. deep-learning)	0.6601

**Table 7 viruses-17-01272-t007:** CCCLin, upper and lower limits for the Aspect ratio calculated for each mitochondria group.

Aspect Ratio					
Dataset	SARS-CoV-2			ZIKV	
	Delta (B.1.617.2)	B.1.1.519	Control	Infected	Control
**CCC Lin**	0.81	0.95	0.92	0.97	0.97
**Lower limit**	0.72	0.94	0.90	0.96	0.97
**Upper limit**	0.87	0.96	0.94	0.98	0.98

**Table 8 viruses-17-01272-t008:** CCCLin, upper and lower limits for Roundness calculated for each mitochondria group.

Roundness					
Dataset	SARS-CoV-2			ZIKV	
	Delta (B.1.617.2)	B.1.1.519	Control	Infected	Control
**CCC Lin**	0.92	0.95	0.93	0.97	0.96
**Lower limit**	0.87	0.93	0.91	0.95	0.95
**Upper limit**	0.95	0.96	0.95	0.98	0.97

**Table 9 viruses-17-01272-t009:** CCCLin, upper and lower limits for Focus calculated for each mitochondria group.

Focus					
Dataset	SARS-CoV-2			ZIKV	
	Delta (B.1.617.2)	B.1.1.519	Control	Infected	Control
**CCC Lin**	0.96	0.97	0.97	0.97	0.96
**Lower limit**	0.93	0.96	0.96	0.96	0.94
**Upper limit**	0.97	0.98	0.97	0.98	0.97

## Data Availability

The original contributions presented in this study are included in the article. Further inquiries can be directed to the corresponding authors.

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
