# Peer review of "Deep Learning-Based Automatic Segmentation and Analysis of Mitochondrial Damage by Zika Virus and SARS-CoV-2"

_viruses, 2025, doi:10.3390/v17091272_

Round 1
Reviewer 1 Report
Comments and Suggestions for Authors
See attached file

Author Response
We greatly appreciate the reviewer’s efforts to carefully review the paper and the valuable suggestions offered. With respect to the concern raised by the reviewer:
- For a reader not familiar with neural networks, the first paragraph in section 2.4 and figure 2 did not make much sense. Perhaps it would help if the authors could point to representative stages in the process, either in a paper or a supplemental figure, to illustrate how it works. Also, the authors should place the work in context: how does the concept of their approach compare with the many other programs now available for automated segmentation of mitochondria in 2-D and 3-D EM data (one is mentioned)?
We thank the reviewer for this suggestion. To improve clarity for readers less familiar with neural networks, we have revised the first paragraph of Section 2.4 to include a more accessible description of the process (lines 186-200). Furthermore, we have expanded the discussion to place our approach within the context of existing methods for automated segmentation of mitochondria in EM images, highlighting similarities and differences with previously reported approaches and emphasizing the advantages of incorporating mitochondria with altered morphologies into the training process (lines 460-470).
- Change ZIKV to SARS-CoV-2 in line 2 of caption of Figure 7. Also indicate explicitly which variants are in the top and bottom rows.
We thank the reviewer for bringing this error to our attention. We have corrected the caption of Figure 7 by replacing “ZIKV” with “SARS-CoV-2” and have explicitly indicated which variants are shown in the top and bottom rows.
- I do not understand why consistency between some manually and automatically extracted parameters was affected by magnification in some cases. Could this be explained by differences in the total number of profiles measured (N) in these cases?
Thanks for this observation. The number of available TEM micrographs and mitochondria differed among the groups (Delta (B.1.617.2), B.1.1.519, ZIKV, and controls). Thus, there are variations in the number of mitochondria segmented and measured at each magnification. Thus, it is possible that this variability could account for the differences observed, rather than an inconsistency in the segmentation method.
- While the “average” correlation coefficient between manual and automated parameters is > 0.9, the correlation for the aspect ratio for the Delta variant of SARS-CoV-2 is considerably lower (0.72-0.87). Is there anything different about those images?
The lower correlation in Aspect Ratio (AR) measurements for the Delta (B.1.617.2) variant could be attributed to the more aggressive or damaging pathology of this variant, which likely alters mitochondrial morphology more significantly and thus impairs automated detection.
SARS-CoV-2 Delta (B.1.617.2) has been consistently associated with more severe clinical outcomes compared to other SARS-CoV-2 variants. Multiple cohort studies have demonstrated that infection with the Delta (B.1.617.2) variant increases the severity of disease compared to B. 1.1.7 [1,2,3]. This suggests a higher burden of cellular and subcellular stress—including mitochondrial damage—which could reduce the consistency between manual and automated measurements. Therefore, it is likely that the diminished correlation observed for AR likely reflects variant-specific mitochondrial disruption in Delta (B.1.617.2) (lines 552-562).
[44] Hyams, C., Challen, R., Marlow, R., Nguyen, J., Begier, E., Southern, J., King, J., Morley, A., Kinney, J., Clout, M., Oliver, J., Gray, S., Ellsbury, G., Maskell, N., Jodar, L., Gessner, B., McLaughlin, J., Danon, L., Finn, A., & AvonCAP Research Group. (2023). Severity of Omicron (B.1.1.529) and Delta (B.1.617.2) SARS-CoV-2 infection among hospitalised adults: A prospective cohort study in Bristol, United Kingdom. The Lancet Regional Health. Europe, 25(100556), 100556.
https://doi.org/10.1016/j.lanepe.2022.100556
[45] Sean P Harrigan, James Wilton, Mei Chong, Younathan Abdia, Hector Velasquez Garcia, Caren Rose, Marsha Taylor, Sharmistha Mishra, Beate Sander, Linda Hoang, John Tyson, Mel Krajden, Natalie Prystajecky, Naveed Z Janjua, Hind Sbihi (2023). Clinical Severity of Severe Acute Respiratory Syndrome Coronavirus 2 Omicron Variant Relative to Delta in British Columbia, Canada: A Retrospective Analysis of Whole-Genome Sequenced Cases, Clinical Infectious Diseases, Volume 76, Issue 3, Pages e18–e25.
https://doi.org/10.1093/cid/ciac705
[46] Ellis, L. P., Hess, O., Huynh, K. L. A., Bearman, G., Kang, L., & Doern, C. D. (2023). A comparison of severity of illness between the SARS-CoV-2 Omicron variant and Delta variant. Antimicrobial Stewardship & Healthcare Epidemiology, 3(1), e188.
doi:10.1017/ash.2023.453
- For future consideration: Since mitochondrial activities are affected by parameters related to the extent of inner membrane folding (variations in crista density and even total loss of cristae), the authors should consider how this feature might be captured, e.g., by a metric for local “texture” perhaps.
This is a very interesting suggestion. In fact, we are working on a similar idea for a more comprehensive study that uses other parameters reflecting inner membrane folding and cristae integrity. This future work explores texture- and edge-based metrics to capture alterations in both the outer membrane and the cristae more effectively. However, including these metrics in this article is outside the scope of our study, as they will require biochemical and immunochemical approaches for validation.
Reviewer 2 Report
Comments and Suggestions for Authors
The authors present an implementation of the U-net convolutional neural network for automated detection, segmentation and measurement of relevant metrics of mitochondria in scanning TEM images of cultured cells. Automated results obtained were compared with manually segmented measurements, and performance metrics were obtained. These aspects were well-presented and discussed. However, there were a number of points of concern that I would like to see addressed:
- In section 2.2 the exclusion criteria for images could be more clearly defined, and whether these were performed manually or in automated fashion should be explained. It is also not clear whether this applies to only the training dataset or to all images used in this study. Given that much of the rationale for a deep-learning image segmentation system is that it performs the tasks automatically, I would like to see the inclusion of this manual step in the workflow addressed directly, potentially with outlined plans to automate this step in future studies.
- My primary concern with this manuscript is that the actual 3D structure of the mitochondrial network is not addressed. Although the system seems to perform well at recognizing mitochondria in TEM images (not a small accomplishment), the extraction of metrics from these mitochondria needs to be carefully considered in the context of the shape of the mitochondrial network. That is, 2D slices through a highly convoluted tubular network can result in a wide range of observed shapes, depending on the size and angle of the slice relative to parts of the network that are intersected. The authors regularly refer to an individual "mitochondrion" when what is being segmented by the algorithm in the image is a slice through a network of tubular structures.
- The biological relevance of the utilized metrics should be more thoroughly discussed, specifically with reference to how mitochondrial fission or fusion would affect those metrics. In particular, this discussion should include a thorough discussion of likely artifacts in the images and derived metrics that are a result of imaging very thin 2D slices of networked 3D structures.
- In relation to the previous point, it would be of value to validate the changes in mitochondria observed by this approach by using an parallel approach to measure mitochondrial changes such as fluorescence microscopy. Even if the fluorescence analysis was performed manually, it would support the value of the system. Given that these studies were performed in cell culture it would be appropriate to perform infections in parallel and use a well-validated fluorescence microscopy approach to stain mitochondria, collect 3D image stacks and assess mitochondrial morphology. Alternatively, a 3D stack of the mitochondrial network could also be obtained with ablative TEM followed by 3D reconstruction.
Author Response
We greatly appreciate the reviewer’s efforts to carefully review the paper and the valuable suggestions offered. With respect to the concern raised by the reviewer:
- In section 2.2 the exclusion criteria for images could be more clearly defined, and whether these were performed manually or in automated fashion should be explained. It is also not clear whether this applies to only the training dataset or to all images used in this study. Given that much of the rationale for a deep-learning image segmentation system is that it performs the tasks automatically, I would like to see the inclusion of this manual step in the workflow addressed directly, potentially with outlined plans to automate this step in future studies.
We thank the reviewer for this comment. In Section 2.2, we have now clarified that image exclusion was performed manually by an expert observer and that these criteria were applied to all datasets used in the study (line 146). We have also acknowledged that, although this manual step was necessary in the present work to ensure high-quality input data, in future studies, we aim to establish a standardized TEM image acquisition protocol to minimize the need for manual exclusion and to ensure more homogeneous datasets for deep-learning training (lines 153-156).
- My primary concern with this manuscript is that the actual 3D structure of the mitochondrial network is not addressed. Although the system seems to perform well at recognizing mitochondria in TEM images (not a small accomplishment), the extraction of metrics from these mitochondria needs to be carefully considered in the context of the shape of the mitochondrial network. That is, 2D slices through a highly convoluted tubular network can result in a wide range of observed shapes, depending on the size and angle of the slice relative to parts of the network that are intersected. The authors regularly refer to an individual "mitochondrion" when what is being segmented by the algorithm in the image is a slice through a network of tubular structures.
We agree that mitochondria constitute a highly interconnected 3D tubular network and that 2D TEM slices inevitably capture only cross-sections of this structure. In the present study, we refer to each segmented element as a “mitochondrion” because the analysis corresponds to the 2D profile of these organelles in thin sections. More importantly, we are focusing on these 2D images because a standard classification system of mitochondrial damage [36] exists, based on thin-section, positive-stained TEM images. In fact, in a previous study, we compared TEM and PCR-based analysis to understand mitochondrial damage [9], finding a very strong correlation between the levels of expression of genes associated with mitochondrial damage and 2D analysis of the mitochondria. Unfortunately, that analysis took us about two years. The goal of this algorithm is to reduce data analysis time from years to weeks, enabling the inclusion of complex architecture and more data through the additional layer of robustness that training deep-learning systems can provide. This approach can use a large number of mitochondria, which are randomly oriented in the cell, and thus allows us to explore a much larger number of orientations. As the reviewer pointed out, having 2D sections represents just a small representation of the organelle architecture; nonetheless, by characterizing hundreds of profiles, we can explore a large universe of conformations and structures. However, we are not aware of any other AI-based system that enables the identification and segmentation of 2D slices of mitochondria from virus-infected cells, where the structural integrity of this organelle is significantly compromised.
[9] Lara-Hernandez, I., Muñoz-Escalante, J. C., Bernal-Silva, S., Noyola, D. E., Wong-Chew, R. M., Comas-García, A., & Co-mas-Garcia, M. Ultrastructural and functional characterization of mitochondrial dynamics induced by human respiratory syncytial virus infection in HEp-2 cells. Viruses, 2023, 15(7), 1518. https://doi.org/10.3390/v15071518
[36] Joshi, M. S., Crouser, E. D., Julian, M. W., Schanbacher, B. L., & Bauer, J. A. (2000). Digital imaging analysis for the study of endotoxin-induced mitochondrial ultrastructure injury. Analytical Cellular Pathology: The Journal of the European Society for Analytical Cellular Pathology, 21(1), 41–48.
- The biological relevance of the utilized metrics should be more thoroughly discussed, specifically with reference to how mitochondrial fission or fusion would affect those metrics. In particular, this discussion should include a thorough discussion of likely artifacts in the images and derived metrics that are a result of imaging very thin 2D slices of networked 3D structures.
We agree with the reviewer; however, to the best of our knowledge, there are no systematic studies that correlate these metrics and artifacts due to the use of 2D sections. As mentioned in the previous answer, Joshi’s classification is widely used and requires 2D sections; it has been used as a standard so far. The subjectivity of this classification can be influenced by bias in the analysis; therefore, we aimed to develop a tool that automatically segments these organelles and extracts metrics using a large number of mitochondria. We are currently working on a detailed biochemical and TEM-based analysis of mitochondrial damage following viral infection; however, the scope of that study is beyond this manuscript. Nonetheless, in our previous work, Hernandez-Lara, I. et al., we were able to correlate the expression of mRNAs associated with mitochondrial damage and some of these metrics, but with severe limitations on the number of organelles studied. In the present version of the manuscript, we have expanded the Discussion to explicitly address the biological relevance of the morphometric parameters analyzed, as well as the technical considerations related to potential artifacts during sample preparation (lines 512-545). We expect that our automated approach will allow us to analyze a large number of mitochondria and thus more robustly address the reviewer’s concerns in future work.
- In relation to the previous point, it would be of value to validate the changes in mitochondria observed by this approach by using a parallel approach to measure mitochondrial changes such as fluorescence microscopy. Even if the fluorescence analysis was performed manually, it would support the value of the system. Given that these studies were performed in cell culture it would be appropriate to perform infections in parallel and use a well-validated fluorescence microscopy approach to stain mitochondria, collect 3D image stacks and assess mitochondrial morphology. Alternatively, a 3D stack of the mitochondrial network could also be obtained with ablative TEM followed by 3D reconstruction.
We agree that further validation is required. However, those experiments are outside the scope of this article. Two-dimensional analysis of TEM data has been used by us and others to determine mitochondrial damage and identify whether fusion or fission is occurring. Within this framework, we have aimed to develop an automated tool that can segment and classify a large number of mitochondria, thereby achieving sufficient statistical power. Using 3D reconstructions alone allows for the study of only a few mitochondria per cell and is highly challenging to use with TEM without coupling it to a milling process. In fact, the best way to do it is using FIB-SEM. Unfortunately, this technology is costly, and we do not have access to it. More importantly, the goal is to characterize the morphology of a large number of mitochondria so that we can have different “snapshots” of the variability in the structure of this organelle in infected cells, so that we can have an idea if there are structural alterations (e.g., based on Joshi’s system) and/or other morphological changes associated with either fusion or fission (e.g., by characterizing the AR and focus of infected vs mock treated cells). While 3D reconstruction gives high-resolution information, this technique is not ideal to answer the questions we are interested in. Finally, fluorescence microscopy has been used to analyze the presence of mitochondrial markers. However, given the size of this organelle and the resolution limit of fluorescence microscopy ( lamda/2), this approach is not ideal to study changes in morphology.